# Personalized Cancer Vaccines Go Viral: Viral Vectors in the Era of Personalized Immunotherapy of Cancer

**DOI:** 10.3390/ijms242316591

**Published:** 2023-11-22

**Authors:** Laura Seclì, Guido Leoni, Valentino Ruzza, Loredana Siani, Gabriella Cotugno, Elisa Scarselli, Anna Morena D’Alise

**Affiliations:** Nouscom, Via di Castel Romano 100, 00128 Rome, Italy; l.secli@nouscom.com (L.S.); g.leoni@nouscom.com (G.L.); v.ruzza@nouscom.com (V.R.); l.siani@nouscom.com (L.S.); g.cotugno@nouscom.com (G.C.); e.scarselli@nouscom.com (E.S.)

**Keywords:** personalized cancer vaccines, neoantigens, viral vectors, adenovirus, poxvirus

## Abstract

The aim of personalized cancer vaccines is to elicit potent and tumor-specific immune responses against neoantigens specific to each patient and to establish durable immunity, while minimizing the adverse events. Over recent years, there has been a renewed interest in personalized cancer vaccines, primarily due to the advancement of innovative technologies for the identification of neoantigens and novel vaccine delivery platforms. Here, we review the emerging field of personalized cancer vaccination, with a focus on the use of viral vectors as a vaccine platform. The recent advancements in viral vector technology have led to the development of efficient production processes, positioning personalized viral vaccines as one of the preferred technologies. Many clinical trials have shown the feasibility, safety, immunogenicity and, more recently, preliminary evidence of the anti-tumor activity of personalized vaccination, fostering active research in the field, including further clinical trials for different tumor types and in different clinical settings.

## 1. Introduction

Cancer vaccines have undergone a remarkable investigation over recent decades as an immunotherapeutic strategy designed to induce new or reinforce existing cytotoxic CD8 T cell lymphocyte (CTL) responses specific to tumor antigens [1,2]. The aim of therapeutic cancer vaccines is to stimulate the patient’s adaptive T cell immune system against specific tumor antigens to control tumor growth both by eradicating the minimal residual disease and by inducing the regression of established tumors. The core concepts for successful therapeutic vaccination against tumors rely on two key factors: the selection of appropriate tumor antigens and a powerful vaccine platform eliciting high-quality CD8 T cells and CD4 T helper cells and the CTL infiltration of the tumor microenvironment (TME) and durable T cells responses [3] (Figure 1). Unfortunately, until now, only one therapeutic vaccine against cancer—sipuleucel-T—has been approved by the United States Food and Drug Administration (FDA) for the treatment of prostate cancer, extending patients’ survival by only 4 months [4]. The lack of success can be attributed to various factors, including the choice of suboptimal antigens, ineffective adjuvants, and poorly immunogenic vaccine platforms, resulting in insufficient CTLs active in the tumor due to an immunosuppressive tumor microenvironment [5,6]. To overcome this challenge, immunotherapies like immune checkpoint inhibitors (ICIs) have been implemented [6]. ICIs, such as anti-CTLA4, anti-PD1, and anti-PDL1 antibodies, have revolutionized cancer care, significantly improving the response rates. Thus, with the advent of ICIs on one hand, and the renewed investment and innovation in the field of vaccinology (which was also spurred by the COVID-19 pandemic), therapeutic cancer vaccines are now re-emerging as a promising approach to significantly enhance the response rates and improve the survival outcomes, particularly when they are used in combination with ICIs. The recent advancements in the field of cancer vaccines include the development of personalized neoantigen-based vaccines with carefully selected individualized antigens, as well as novel combination approaches to bolster the immune responses [7]. The presence of neoantigens has been shown to correlate with a positive response to immunotherapies, making them appealing targets for personalizing cancer vaccines to enhance their effectiveness.

Out of approximately 400 ongoing clinical trials assessing cancer vaccines, 49 are personalized vaccines (ClinicalTrials.gov, accessed on 15 September 2023). Different vaccine platforms and delivery strategies are currently being tested in clinical studies of personalized vaccination, including peptides, Dendritic Cells (DCs), DNA, RNA, and viral vectors. The key factor in the choice of the most appropriate cancer vaccine platform relates to its capability to elicit an adequate quantity and quality of neoantigen-specific T cell responses and to how rapidly the vaccine can be designed, manufactured, and finally administered to each patient. In this review, we provide an outlook of the most recent clinical results obtained in the field and discuss the key features of viral vectors and their great potential for precision medicine against cancer, with focus on the novel vectors based on the Great Ape-derived Adenoviruses (GAd) and Modified Vaccinia Virus Ankara (MVA) vectors.

## 2. Tumor Antigens

The selection of the optimal target antigen is a priority to generate an effective vaccine. Tumor antigens can be classically categorized into tumor-specific antigens (TSAs) arising from cancer mutations, which are exclusively presented on tumor cells, and tumor-associated antigens (TAAs), which can also be present on additional tissues [8]. The vast majority of previous cancer vaccines have targeted TAAs, which are self-antigens that include proteins overexpressed in cancer cells (e.g., HER2, hTERT, and mesothelin), differentiation antigens, mainly expressed on the tissue from which the tumor originated (e.g., gp100, tyrosinase, and Melan-A/MART-1), and cancer germline/cancer testis antigens normally expressed only in male germ cells, placentas, or in the early stages of embryonic development (e.g., NY-ESO-1, MAGE-A1, and MAGE-A3) [9]. The main hurdles associated with the use of TAAs as vaccine target antigens are the immunological tolerance established against autologous proteins and the risk of undesirable autoimmunity due to the expression of TAAs in normal tissue. Differently from TAAs, TSAs or neoantigens generated by mutations in cancer cells are tumor-specific and not subjected to central immune tolerance, but rather recognized as foreign by the immune system. Therefore, compared to TAAs, neoantigens can induce stronger anti-tumor immunity. Moreover, cellular immunity directed against neoantigens is specific to the mutant peptides and has been shown not to cross-react with wild-type antigens [10], suggesting that the targeting of neoantigens is safe, with minimal risk of damage to the healthy tissues and off-target effects. 

Insertions or deletions (indels), point mutations, and gene fusions are typical mutations that lead to the generation of neoantigens [11], with the single-nucleotide variants (SNVs) representing a major source of neoantigens. Most neoantigens are patient-specific; indeed, the majority of cancer somatic mutations giving rise to neoantigens are stochastically generated during cell duplication. As a consequence, every patient acquires their own unique set of mutations, mandating a personalized tailored-made intervention. Public or shared neoantigens derived from driver mutations in oncogenes or other hotspot mutations across the genome are also attractive therapeutic targets, but they represent an “elite” class of shared cancer-specific epitopes present in a subset of patients with a given tumor subtype [12].

The frequency of mutations measured using the tumor mutation burden (TMB) is defined as the total number of substitutions, insertions, and deletions per megabase in the exon coding region. Independent studies conducted across various cancer types and immunotherapy treatments have consistently found that a higher TMB is associated with improved clinical outcomes, likely due to the increased potential for generating a higher number of neoantigens that can trigger a robust immune response against cancer cells; however, this association may not be consistent across all cancer types [13,14].

The anti-tumor efficacy relies not only on the quantity of neoantigens, but also on their attributes, such as their immunogenicity, clonality, and expression by tumor cells. Indeed, in addition to the amount of a presented antigen, the distribution of an antigen across cancer cells, known as clonality, is of paramount importance. The sub-clonal antigens present in only a fraction of tumor cells can potentially facilitate tumor escape by promoting the outgrowth of antigen-deficient cells. The success of immune control across all sites of metastatic disease by targeting sub-clonal antigens is less likely. Notably, the burden and fraction of clonal neoantigens correlate with the response to ICIs in lung cancer and melanoma [15]. Recurrent neoantigens, which are functionally relevant and often clonal, serve as superior targets for therapy. However, even when targeting clonal antigens derived from critical proteins, evasion mechanisms can still occur. For this reason, approaches targeting multiple antigens simultaneously or sequentially is considered to be more effective [15,16].

## 3. High-Throughput Identification of Neoantigens for Personalized Cancer Vaccines

One of the critical aspects in developing personalized cancer vaccines is the accurate identification of tumor-specific and somatic mutations to select the “best” subsets of candidate neoantigens for optimal immune responses [7,17,18,19,20,21]. The wide majority of candidate neoantigens derive from point mutations (SNVs) or small indels. Tumor-specific point mutations are, nowadays, easy to detect with the current Exomeseq Next-Generation Sequencing (NGS) platforms by comparing the DNA extracted from a tumor tissue biopsy with that from a matched “healthy” sample (blood or tumor-adjacent tissue) (Figure 2). Several workflows for somatic variant calling exist and, for most of them, very high precision and recall rates have been reported especially to detect somatic variants with a medium/high allele frequency [22]. 

The advancement of NGS analytical methods has underscored that the total number of somatic mutations and corresponding candidate neoantigens varies significantly with tumor histology and among individual patients. Tumors with a high TMB, such as melanoma and Non-Small-Cell Lung Cancer (NSCLC), encode hundreds of potential neoantigens [23]; however, only a very limited fraction of these mutations (1–2%) holds the capacity to efficiently stimulate the immune system against tumor cells [21]. Since the majority of vaccination platforms currently under development have a limited capability of encoding mutated peptides, it is impossible to target all or even most of the candidate neoantigens detected in a patient. Therefore, there is a need for efficient methods that prioritize the mutated neopeptides to select those that have the highest likelihood of being neoantigens involved in tumor rejection. 

In order to be effective, vaccine targeted neoantigens need to be abundant, possibly encoded by a large number of tumor cells (clonal), well exposed by the Major Histocompatibility Complex classes I (MHC I) and II (MHC II), and find the cognate T cell receptor (TCR) [24]. Nowadays, the neopeptides abundance and the chance of binding to MHC can be determined with very high precision. The availability of large tumor proteogenomic Mass Spectrometry data has facilitated the enhancement of machine learning-based predictors. Currently, it is possible to estimate the probability of binding to MHC I and, to a lesser degree of accuracy, binding to MHC II. In contrast, predicting the liability of recognition by a T cell remains a challenge due to the complexity of the physical mechanisms governing the binding of TCRs to peptides once they are complexed with the MHC protein [25]. The currently available neoantigen prioritization methods can be categorized into three macro-categories based on the manner in which the aforementioned parameters are used to rank the lists of candidate neopeptides. 

The first group of multiparametric filtering methods applies user-specified filters in order to select the candidate neopeptides that satisfy all the defined criteria. The representative tools in this category are pVACTools [26], TIminer [27], and Progeo-Neo [28]. The available filters include variant expression, variant allelic fraction, and the likelihood of neopeptide binding affinity using MHC I and II predictions. The pVACTools also include the possibility to predict stability of MHC–neopeptide complexes by using the software NetMHCStabPan [29]. TIminer includes the possibility to estimate the immunogenicity of the tumor by characterizing the tumor-infiltrating immune cells detected in the analyzed biopsy. The major drawback of these tools is that neoantigen prioritization is based on predetermined thresholds for the parameters of interest. The use of these approaches might result in the selection of a suboptimal number of candidate neoantigens that pass the thresholds due to the heterogeneity of the tumor or the characteristics of the tumor biopsy analyzed to design the vaccine. Therefore, it might be necessary to re-defined them in multiple runs of analyses. This potential back and forth is suboptimal in a clinical scenario, where it is important that the time from collection of the biopsy to the injection of the vaccine is short to avoid changes in the tumor mutation landscape and disease progression occurring prior to vaccination. 

The second group of prioritization methods relies on score ranking and combines different parameters to create a ranked list of potential neoepitopes arising from the input variants. The advantage compared to the sequential filtering is the possibility to rank all the neopeptides detected in a tumor biopsy by also including those that are good only for some of the required filtering criteria. Among the representative software in the category, there is VaxRank [30] and VENUS [31]. VaxRank incorporates metrics in its scoring system that consider the variant-specific expression based on the supporting RNA-Seq read counts determined in a tumor biopsy. In addition to this, VENUS combines the abundance of the transcripts carrying the mutation with the allele frequency of the mutation and the likelihood of the MHC I binding of the detected neopeptides, aiming to produce a final rank that better reflects the degree of neopeptide presentation in the immune system. 

The third group of neoantigen prioritization methods considers the probability of interaction between the TCRs and MHC–neopeptide complexes. These methods are still in the exploratory stage. The primary limiting factor for the development of more accurate predictors is inherent to the requirement for TCRs to be “plastic” in order to bind with a high affinity for a diverse universe of potential antigens, surpassing the number of unique TCRs present in an individual by some orders of magnitude.

TCR cross-reactivity is an essential feature of the immune system, and it is driven by the fact that TCRs focus on “hot spot” regions of the peptide that exhibit structural and chemical similarities among different agonist ligands. Outside of these “hot spots”, more sequence diversity is permitted [25]. A representative method of this group is the neoantigen quality model that has been used to characterize the extent of immunoediting in long-term survivors of pancreatic cancer [32]. The neoantigen quality model is based on the concept that a neoantigen can be immunogenic if there is a TCR capable of discriminating it from its wild-type counterpart. The estimation of immunogenicity is determined by two features: the differential MHC presentation, which is determined as the ratio of the predicted MHC binding affinity between mutant and wild-type neopeptide, and differential T cell reactivity. The latter is determined by estimating whether the amino acid(s) variation could enhance the binding strength with the TCR, which is assessed with an experimental affinity dataset. 

The development of neoantigens prioritization methods is a very active area of research continuously boosted by the novel validated neoantigens detected in patients vaccinated in ongoing clinical trials. The increasing availability of these data will certainly allow the future development of new, better tools. 

## 4. Viral-Vectored Vaccines Targeting Tumor Neoantigens

A number of different platforms are currently being evaluated in clinical trials of neoantigen-based vaccines, including peptides, DC, DNA, RNA, and viral vectors. The induction of robust T cell immunity is key to developing an effective cancer vaccine. More specifically, cytotoxic CD8 T cells are the most relevant effectors for an effective anticancer immune response. For the induction of cytotoxic T cells, it is important that the antigen is expressed within the cell in order to be processed with the proteasome machinery and properly loaded onto MHC I complex. In this context, recombinant viral vectors are among the preferred technologies due to their ability of triggering powerful and long-lasting cellular responses. Genetically modified viruses, such as adenovirus, parvovirus, vaccinia virus, lentivirus and adeno associated virus (for these two latter ones, there are no current clinical trials of personalized vaccines), have emerged as favorable carriers for encoding and delivering neoantigens. They act as “viruses” and efficiently transduce cells, including professional Antigen-Presenting Cells (APCs), expressing high levels of the transgene, and ultimately leading to a potent immune response by the CD4 and CD8 T cells [33]. Numerous studies have shown that the use of viral vectors for antigen delivery results in enhanced immunogenicity compared to that of the other platforms, likely due to the pro-inflammatory environment induced by viral protein expression [34,35]. Indeed, viral vectors can induce high-level immunogenicity without the use of exogenous adjuvants. Their intrinsic adjuvant properties are due to the expression of diverse Pathogen-Associated Molecular Patterns (PAMPs) and the activation of innate immunity. Among the most comprehensively studied vaccine viral vectors are Adenoviruses (Ads) and MVA, an attenuated vaccinia virus.

Adenoviruses are non-enveloped, double-stranded DNA viruses with a genome of 36 kbp that can accommodate cDNA sequences of up to 7.5 kbp. Genome replication takes place in the nucleus, but remains extrachromosomal, minimizing the risk of insertional mutagenesis associated with this vector. The adenoviral vectors commonly used for vaccination are replication-incompetent through the deletion of the E1 region, which limits their pathogenicity, while still enabling the generation of humoral and cellular responses to transgenes. Based on these features, Adenoviruses display several advantages as viral vectors for vaccine development: low-level pathogenicity, genetic safety, a lack of integration in the host genome, strong immunogenicity, the efficient infection of different cell types, a high transgene incorporation capacity, and the relative ease of vector construction and production under Good Manufacturing Practice (GMP). Human Adenovirus serotype 5 (Ad5) is the most widely used vector, and Ad5-based vaccines have been shown to induce potent T cell responses [36,37]. However, a pre-existing immunity to Ad5 can significantly blunt the immune responses induced by Ad5-vectored vaccines in humans [36], limiting its clinical application. The issue of a pre-existing immunity can be circumvented by the use of vectors isolated from different species, such as Great Ape-derived Adenoviruses (Gad) [38]. GAd is still similar to human Ads, but markedly differs in its antigenic determinants particularly in its hexon Hyper Variable Regions (HVRs) and fiber protein domains, leading to a notably less seroprevalence in human populations worldwide [39]. Currently, two GAd vectors are in clinical development as cancer vaccines: a Chimpanzee Adenovirus (species E ChAd68 serotype) and a Gorilla Adenovirus (species C GAd20 serotype), which have both been proven to be capable of inducting optimal T cell immunity in cancer patients [40,41,42].

Poxviruses have a long and successful history in vaccination programs and are one of the first and most commonly used class of viral vectors for heterologous gene delivery. Poxviruses are a double-stranded DNA virus with a linear genome. They have the ability to accept large inserts of foreign DNA, over 24,000 bp, and attenuated strains like MVA were shown to efficiently and stably express recombinant antigens from an expression cassette of more than 7.5 kb. MVA was originally generated by serial passages of Chorioallantois Vaccinia virus Ankara in chicken embryo fibroblast cells (CEF). The serial passages (more than 500) of the virus in CEF cultures resulted in major deletions and mutations in the viral genome, with a loss of about 30 kbp (i.e., 15% genome loss). These modifications resulted in a highly attenuated vaccine virus. The virus is unable to replicate efficiently in human and most other mammalian cells [43], but the replication defect occurs at a late stage of virion assembly, allowing unimpaired viral and recombinant gene expression, making MVA an efficient single round expression vector incapable of causing infection in mammals. Viral replication and transcription occur exclusively in the host cell’s cytoplasm, eliminating the risk of insertional mutagenesis [38]. MVA has been administered to over 120,000 humans during the smallpox eradication campaign, with an excellent safety record. Its safety has been demonstrated in a number of pre-clinical and clinical studies [44,45,46,47,48], rendering MVA a leading candidate vaccine vector against a spectrum of infectious diseases and cancers. Especially in the context of vaccine regimens based on heterologous prime-boost, MVA has exhibited exceptional capability as a booster vector. Moreover, MVA-based vaccines benefit from a robust track record of design and manufacturing. 

The capability of these viral vectors to target multiple neoantigens simultaneously is key to addressing tumor antigen heterogeneity, allowing the targeting of both dominant and subdominant clones, and to potentially curtailing tumor immune escape. In addition, this approach of polyepitope neoantigen vaccine offers the advantage of possibly overcoming the current limitations of the prediction algorithms, increasing the likelihood to encode in the vaccine at least a subset of neoantigens eliciting effective tumor-specific T cell responses post vaccination.

A primary limitation of vaccination strategies reliant on viral vectors is the development of anti-vector immunity following the initial immunization, leading to inefficient boosting upon the re-administration of the same vector: the so called “homologous prime–boost”. In this respect, MVA vectors are less sensitive to prior anti-vector immunity and therefore can be re-administered several times without affecting their ability to further improve the antigen-specific immune responses. 

Clinical studies have consistently demonstrated that a heterologous prime-boost approach [38,42,49,50] involving different platforms can elicit stronger immune responses compared to those of repeated vaccinations with a single viral vector. The combination of GAd prime with MVA boost is among the most extensively studied heterologous prime-boost approach, and it has been shown to induce exceptionally strong T cell responses in number of different infectious disease trials and using different transgenes [45,47,51]. In the context of cancer vaccine, the heterologous ChAdOx1-MVA vaccination regimen targeting the oncofetal self-antigen 5T4 has been tested in early-stage prostate cancers and found to be safe and immunogenic, with the detection of ex vivo T cell responses to the vaccine-encoded tumor-associated antigen 5T4 [50].

In the context of personalized neoantigen vaccines, two major technologies are currently used to boost GAd primed T cell immunity; one is based on self-amplifying RNA [40], and the second one on MVA vectors [41]. 

Our group is developing a personalized vaccine based on GAd prime, followed by the MVA vector encoding 60 unique patient-specific neoantigens (NOUS-PEV, discussed next), which is currently employed in a clinical trial on patients with metastatic melanoma and NSCLC in combination with pembrolizumab (NCT04990479). 

## 5. Manufacturing Viral Vector-Based Personalized Cancer Vaccines

The key challenge in the clinical application of personalized vaccines lies in the necessity for the rapid manufacturing and timely delivery of individually tailored vaccines for patients with advanced diseases.

The manufacturing of personalized cancer vaccines presents distinct complexities depending on the delivery platform, e.g., whether it is a peptide, DCs, DNA, RNA, or viral vector. 

From a regulatory perspective, viral vectors personalized cancer vaccines fall under the regulatory umbrella of Advanced Therapy Medicinal Products (ATMPs), a field that, due to a rapid technological development, confronts several challenges associated with manufacturing activities [52]. However, the regulators who are aware of the complexity, but also of the vast potentiality behind such cutting-edge technologies, are willing to accept a certain degree of flexibility in laying down the GMP requirements applicable to personalized ATMPs, as long as the product attributes related to quality, safety, and efficacy are assured [53]. The instance arises when, due to some long-lasting tests like sterility (Ph. Eur. 2.6.1), it is not feasible to fully release the finished product, as it must be administered immediately after manufacturing. In this scenario, the regulators accept a two-step release procedure, in which time-consuming tests are completed post injection, provided that an adequate safety control strategy is designed [53]. 

CAR T cells led the way in the field of personalized therapies on the quality controls and key tests for the final product release [52]. The quantity, purity, and potency are critical quality attributes (CQAs) of the manufacturing process. Due to the variability in the personalized starting material, the performance of the manufacturing process can be suboptimal, affecting the yields, clearance (purity), and potency, independently from the specific platform. The variability in the starting material is a crucial aspect that is always considered in the definition of the products’ release criteria. Based on the current data, the failure rate of the manufacturing process for each specific platform has not yet been established.

The viral-vectored neoantigen vaccines currently in clinical development include NOUS-PEV (Nouscom), GRANITE (Gritstone Bio), and myvac TG4050 (Transgene) [54]. Essentially, these viral vector delivery platforms rely on the utilization of replicative defective Adenoviruses and/or MVA, and the manufacturing workflow is reported in Figure 3.

The manufacturing performance of personalized viral vector vaccines is assessed using the Turn Around Time (TAT), which is the duration from the tumor biopsy to the final product release. For the NOUS-PEV vaccine, which employs GAd priming, followed by MVA boosting, the reported TATs are approximately 8 weeks for the GAd vector, marking as one of the swiftest TATs thus far, and approximately 11 weeks for MVA. The longer production time for MVA is compatible with its delivery as a booster vector 3 weeks afterGAd administration. In the GRANITE trial, the Adenovirus, encoding up to 20 tumor-specific neoantigens, was developed and released within a TAT of 14–18 weeks [40]. 

Viral vector personalized vaccines compete with the other personalized ATMP to assure the development of an affordable and commercially viable product. At the moment, the costs associated with the manufacturing and release of personalized ATMP are still too high for a commercially valuable product; therefore, technological investments are ongoing to optimize the manufacturing workflow [54]. Based on promising clinical results, the field is attracting funding from major pharmaceutical companies and private partners, fostering technological development and the optimization of the operational costs [54]. In the field of ATMP, CAR-T cell therapy is the most advanced, with several commercially approved products and many in clinical trials. Therefore, several companies investing in CAR-T manufacturing development identified five innovation drivers that could make these products cheaper and more affordable. Vector and gene engineering innovation, process improvement, hardware innovation, digital innovation, and point-of-care manufacturing are the five key points that could potentially reduce the manufacturing costs of autologous cell therapies by up to 75% [55]. Likewise, viral vector, personalized vaccines should stay abreast and leverage the CAR-T cell experience, working on the same innovation drivers to achieve a comparable reduction in the manufacturing-related costs. Finally, the success of viral vector personalized vaccines cannot solely rely on process and cost optimization. 

A modern healthcare system and continuous support from regulatory agencies must guide the growth and maturation of cancer personalized therapies, with the ultimate goal to achieve a comparable reduction in the manufacturing-related costs [54].

## 6. Clinical Trial Landscape of Personalized Vaccines 

Numerous clinical trials are attempting to induce tumor control by using personalized neoantigen-based vaccination, particularly in the setting of solid tumors with a high neoantigen load, with NSCLC and melanoma the most common indications, but also in “cold tumors” with low mutation burden, such as Microsatellite Stability (MSS) tumors and Pancreatic Ductal Adenocarcinoma (PDAC) [40,56,57,58,59]. Table 1 presents a summary of the industry-sponsored clinical trials based on different platform technologies encoding tumor neoantigens, which are often tested in combination with PD1, PDL1, and/or CTLA4 inhibitors in various tumor types.

Multi-neoepitope approaches are typically pursued, with the number of targeted neoantigens ranging from five to sixty depending from the vaccine platform, with the highest payload being delivered by the viral vectors [38]. Overall, the first-in-human clinical trials of personalized cancer vaccines have demonstrated the feasibility, safety, and capability to elicit neoantigen-specific T cell responses, although to a different extent in relation to the type of vaccine used, and promising preliminary results on clinical activity. 

The excitement surrounding personalized cancer vaccines surged after the release of the first efficacy data in the field from a randomized phase II trial of Moderna’s personalized vaccine mRNA-4157 [60,61,62] plus pembrolizumab versus pembrolizumab alone as an adjuvant therapy in high-risk melanoma patients (NCT03897881). More specifically, this phase IIb trial enrolled 157 patients with stage III/IV melanoma. Following complete surgical resection, the patients were randomized to receive mRNA-4157 (nine total doses) and pembrolizumab (n = 107) versus pembrolizumab alone (n = 50) [62]. No dose-limiting toxicities or grade 3–4 adverse events were observed, and the patients treated with the combination of mRNA-4157 and pembrolizumab had a 44% higher rate of recurrence-free survival than that of the patients treated with pembrolizumab alone. Based on these results, the FDA granted a breakthrough therapy designation to mRNA-4157 in combination with pembrolizumab for the adjuvant treatment of patients with high-risk melanoma [63]. 

Promising results came also from a recent phase I study from Biontech in patients with surgically resected PDAC tumors treated with RO7198457 mRNA vaccine encoding 20 neoantigens in combination with atezolizumab and mFOLFIRINOX. The elicitation of neoantigen-specific T cell responses was shown in 50% of patients (eight out sixteen), with an intriguing correlation between vaccine-induced neoantigen-specific immunity and delayed disease recurrence in those patients with a positive immune response [59]. It is interesting to note that the anti-tumor activity of RO7198475 mRNA in combination with atezolizumab in patients with advanced-stage solid tumors (NCT03289962) has been found to be modest, suggesting that the administration of personalized vaccines in an adjuvant setting or earlier-stage diseases may be an ideal clinical setting in which to apply these vaccination approaches. Indeed, in a disease-free state, there is less immune suppression associated with the tumor and a more favorable environment for the immune system to mount effective T cells responses. 

In such an enthusiastic era of personalized vaccines, viral vectors are also emerging as a promising approach to anti-cancer therapies, given their key features (discussed in the previous paragraph), such as the type and durability of induced immune responses, the ability to be combined in well-known and extensively validated heterologous prime-boost regimens, the capacity to encode large antigens, and their extremely safe profile. Differently from RNA vaccines, in particular unmodified RNA lipoparticles, the profile of local and systemic reactogenicity post vaccination with vectors such as Adenoviruses and MVA is very favorable. Indeed, the high reactogenicity, including Grade 3 systemic adverse events, linked to the strong induction of innate immunity is induced by RNA intravenous injection in humans [64]. 

Among the viral vector-neoantigen based vaccine platforms, there is Gritstone bio’s personalized vaccine, GRANITE, which relies on a heterologous prime-boost approach of ChAd68 prime and self-amplifying RNA boost. The results from a phase I/II study on patients with advanced metastatic solid tumors (NCT03639714) have shown the robust and consistent induction of CD8 T cells against multiple neoantigens and IFNγ responses, detectable via ex vivo ELISpot in 100% of the patients upon heterologous prime-boost vaccination [40]. Early signals of clinical efficacy in several patients with metastatic Microsatellite Stable Colorectal Cancers (MSS-CRC) have also been reported.

The Nouscom personalized vaccine, NOUS-PEV, is a vaccine designed to target 60 patient-specific neoantigens. Vaccination is based on GAd vaccine prime, followed by MVA boost. In a phase I clinical trial (NCT04990479), NOUS-PEV was administered in combination with pembrolizumab to patients with metastatic malignant melanoma and NSCLC (n = 6). Vaccination was feasible and safe, and immunogenicity was demonstrated in all the evaluable patients who received the prime-boost regimen, with the detection of robust neoantigen-specific immune responses to multiple neoantigens in the peripheral blood comprising both CD4 and CD8 T cells [41]. The expansion and diversification of vaccine-induced TCR clonotypes were observed in the post-treatment biopsies of the patients with clinical response, providing evidence for the homing and infiltration of neoantigen-induced T cells into the tumor [41]. A similar approach based on GAd/MVA (NOUS-209) is also currently employed to target shared neoantigens in metastatic patients with a deficiency of dMMR/MSI. The results from a phase I clinical trial (NCT04041310) combining the NOUS-209 vaccine with anti-PD1 antibody pembrolizumab in patients with metastatic colorectal, gastric, and gastro-esophageal cancers have been recently published [42]. The combination treatment was reported to be safe, highly immunogenic, and demonstrated promising early signs of clinical efficacy, with no dose-limiting toxicities. NOUS-209 is currently being investigated in multi-center Europe and United States phase II randomized clinical trials on patients with dMMR/MSI-H unresectable and metastatic CRC in combination with ICIs versus ICIs alone. 

## 7. Conclusions and Future Directions

Advancements in sequencing technologies, antigen selection platforms, and combination immunotherapy strategies have brought renewed optimism for personalized cancer vaccines, but many challenges still exist. 

There are several mechanisms responsible for a cancer vaccine failure that can be ascribed into two major categories: vaccine-related factors and tumor-dependent factors. The factors belonging to the first category include the inability of a certain vaccine type to induce robust and long-term T cell immunity, with a good T cell quantity and quality. Indeed, inadequate memory responses can result in tumor recurrence. The selection of optimal antigens is also crucial in this scenario, as it may affect the ability to generate a strong and broad immune response. The current strategies, including the improvement of the current neoantigen prediction algorithms, are in place to increase the likelihood of selecting and targeting the “right” set of neoantigens. The other factors responsible for a vaccine failure, however, are more closely dependent on the tumor microenvironment, with tumor-intrinsic and extrinsic factors playing an important role in suppressing the immune system, thereby impeding the development of a robust and effective immune response [3]. These factors include the expression of immunosuppressive proteins, such as PDL1 and others on tumor cells, the infiltration of immunosuppressive cells (such as Cancer-Associated Fibroblasts, Myeloid-Derived Suppressor Cells, T regulatory cells, and M2 macrophages), the loss of tumor antigen expression, the loss of HLA expression, and the alteration of antigen processing pathways. Better knowledge of the TME and the deeper characterization of predictive biomarkers are key to identify the subgroups of patients who are most likely to derive benefits from cancer vaccination. 

The field is moving forward to tackle critical aspects of this novel personalized intervention. The major goals are to improve the overall vaccine production process to make it faster and cheaper. Given their unique features, viral vectors hold great promise and are expected to have a central role in these upcoming developments.

## Figures and Tables

**Figure 1 ijms-24-16591-f001:**
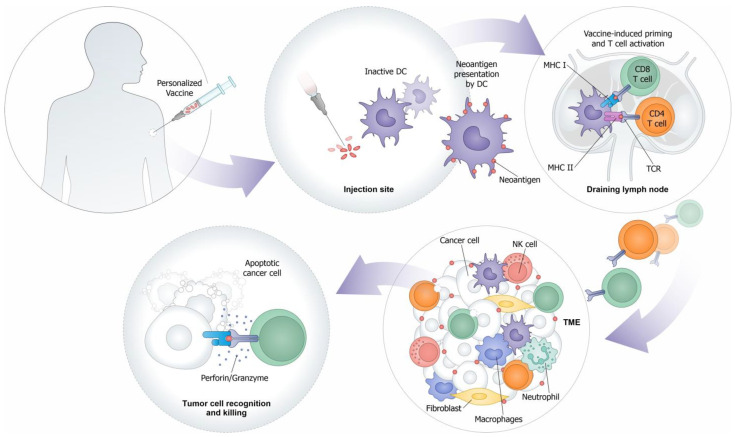
Mechanism of action of personalized cancer vaccine. After vaccination at the injection site, Dendritic Cells (DCs) or other Antigen-Presenting Cells (APCs) are loaded with antigens and migrate to the draining lymph node. There, the APCs present the antigens through the Major Histocompatibility Complex classes I (MHC I) and II (MHC II), respectively, to the CD8 and CD4 T cells. The activated T cells with an antigen-specific T cell receptor (TCR) proliferate and migrate to the tumor microenvironment (TME), where they recognize and kill cancer cells through the release of cytotoxic granules.

**Figure 2 ijms-24-16591-f002:**
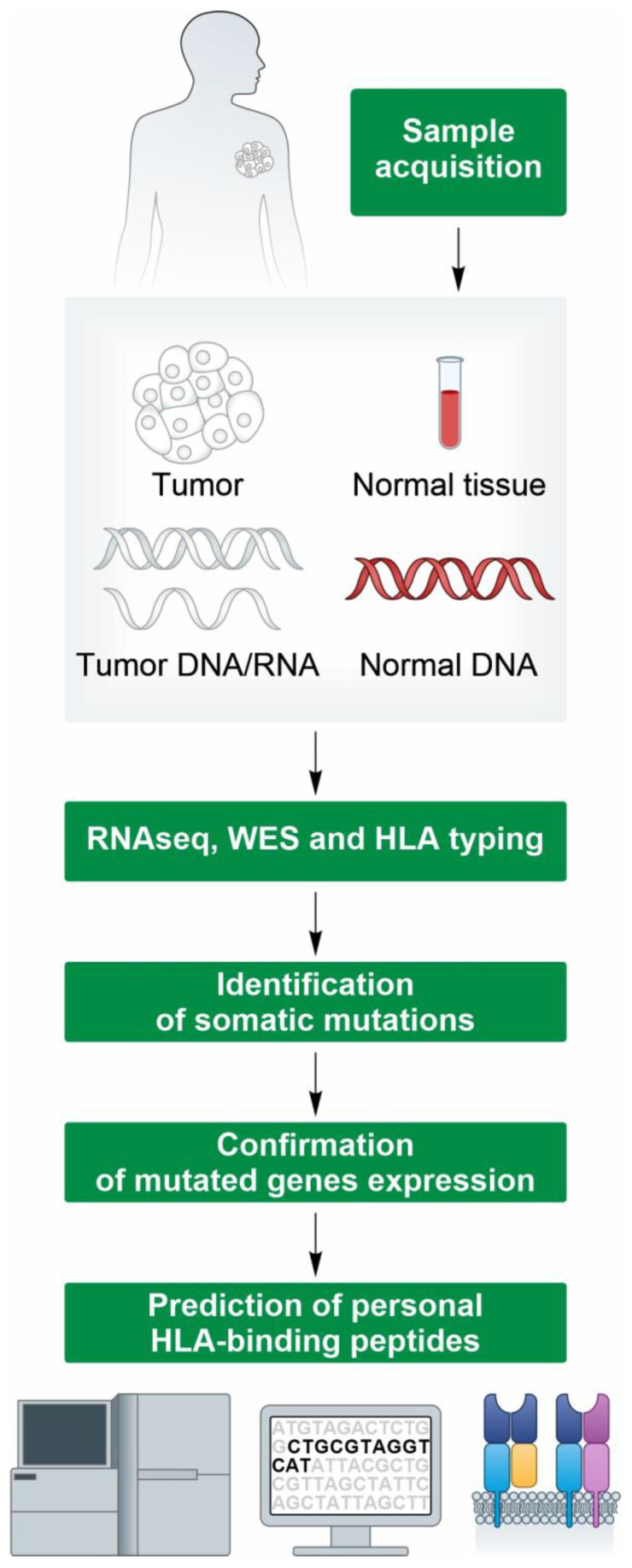
Engineering individualized neoantigen vaccines. The next-generation sequencing of a patient’s healthy tissue (such as peripheral blood mononuclear cells (PBMCs)) and a tumor biopsy sample is performed. The sequenced data from tumor and normal DNA are compared to identify the tumor-specific mutations. The mutations are prioritized as vaccine candidates on the basis of their likelihood to elicit a T cell response via computational methods, such as human leukocyte antigen (HLA)-binding prediction, the quantification of mutated transcript expression, and the clonality of the mutation and other features.

**Figure 3 ijms-24-16591-f003:**
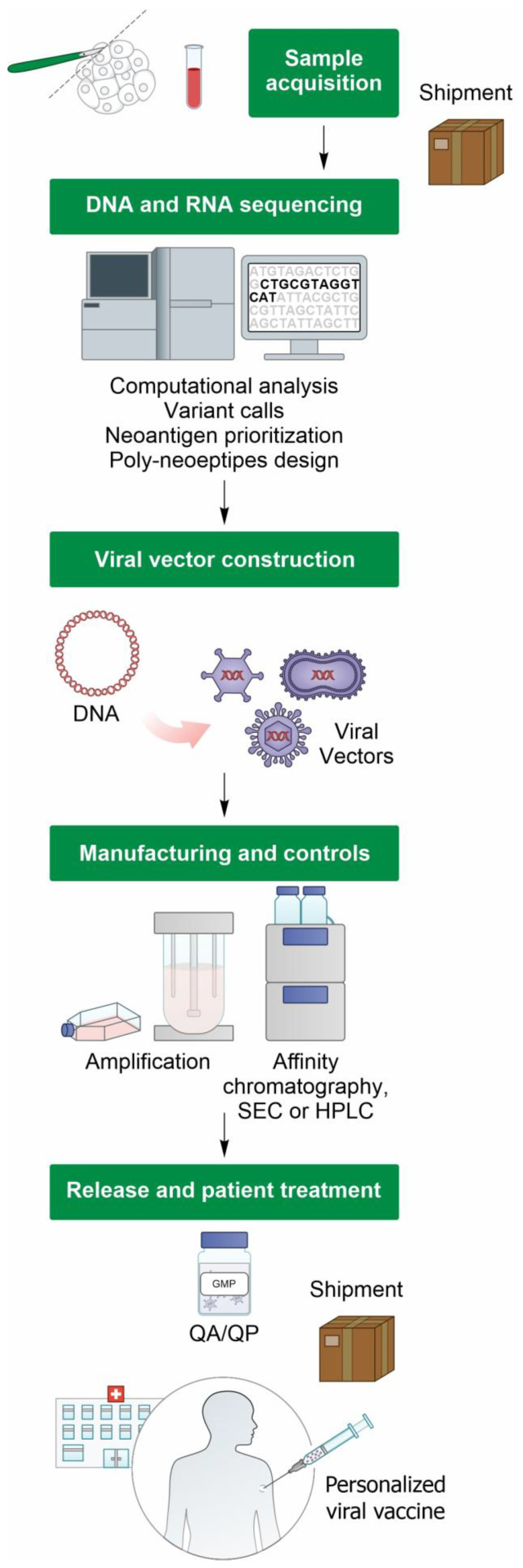
Manufacturing workflow of viral vector-based personalized cancer vaccines. Specimens (tumor and normal) are collected at the clinical site and shipped for next-generation sequencing (NGS) analysis. Subsequently, both the DNA and RNA undergo next-generation sequencing, which is followed by computational analysis, variant calls, neoantigen prioritization, and poly-neoepitope design. A viral vector is then constructed, and small-scale Good Manufacturing Practice (GMP) and analytical control processes are performed, leading to Quality Assurance/Quality person (QA/QP) release. Finally, the drug product is shipped to the clinical site for patient administration.

**Table 1 ijms-24-16591-t001:** Personalized cancer vaccine trials.

Vaccine	Company	Platform	Number of Neoantigens	Phase	Main Indication	Adjuvant	Route of Administration	Trial Number	Status	Immunotherapy	Vaccine Doses
iNEO-Vac-P01	Hangzhou Neoantigen Therapeutic	Peptide	5–20	I	Advanced Pancreatic cancer	GM-CSF	Subcutaneous	NCT03645148	Completed	aPD1	7
					Advanced Pancreatic cancer	GM-CSF		NCT03662815	NA		7
					Pancreatic cancer	GM-CSF		NCT04810910	Recruiting		7
					Esophagus cancer	GM-CSF		NCT05307835	Recruiting		7
EVX-01	Evaxion Biotech/Merck	Peptide	Up to 10	II	Metastatic melanoma		Intramuscular	NCT05309421	Recruiting	aPD1	10
EVX-02	Evaxion Biotech	DNA	Up to 13	II	Advanced melanoma		Intramuscular	NCT04455503	Active, not recruiting	aPD1	8
VB10.NEO	Nykode/Genentech	DNA	Up to 40	I	Advanced solid tumors (melanoma, NSCLC, RCC, UC, HNSCC, TNBC, gastric/GEJ cancer, cervical, anal, or MSI-high tumors)		Intramuscular	NCT05018273	Recruiting	aPDL1	15
				II	Melanoma, NSCLC, RCC, and SCCHN		Intramuscular	NCT03548467	Completed	Bempegaldesleukin-NKTR-214	14
GNOS-PV02	Genos Therapeutics	DNA	Up to 40	II	Metastatic liver cancer	plasmid encoded IL-12 (INO-9012)	Intradermal injection and electroporation	NCT04251117	Active, not recruiting	aPD1	NA
GRANITE	Gritsone bio	Viral DNA ChAd68 and samRNA	Up to 20	II	Colon cancer		Intramuscular	NCT05456165	Terminated	aPDL1 and aCTLA4	GRT-C901 2 doses and GRT-R902 4 doses
					NSCLC, colorectal cancer, gastroesophageal adenocarcinoma, and urothelial carcinoma		Intramuscular	NCT03639714	Completed	aPD1 and aCTLA4	GRT-C901 2 doses and GRT-R902 4 doses
				III	Metastatic colorectal cancer		Intramuscular	NCT05141721	Active, not recruiting	aPD1, aCTLA4, aVEGF	GRT-C901 2 doses and GRT-R902 4 doses
NOUS-PEV	Nouscom	Viarl DNA GAd and viral DNA MVA	Up to 30	I	Metastatic NSCLC and melanoma		Intramuscular	NCT04990479	Active, not recruiting	aPD1	Gad 1 dose and MVA 3 doses
RO7198457	BioNTech/Genentech	Unmodified mRNA	Up to 20	I	Melanoma, NSCLC, bladder cancer, colorectal cancer, TNBC, RCC, and head and neck cancer		Intravenous	NCT03289962	Active, not recruiting	aPDL1	at least 8
				I	Pancreatic cancer		Intravenous	NCT04161755	Active, not recruiting	aPDL1	10
				II	Colon cancer		Intravenous	NCT04486378	Recruiting		15
				III	Advanced melanoma		Intravenous	NCT03815058	Active, not recruiting	aPD1	at least 8
mRNA 4157	Moderna/Merck	Modified mRNA	Up to 34	I	NSCLC, SCLC, melanoma, bladder urothelial carcinoma, HPV-ve HNSCC, and MSI-high tumors		Intramuscular	NCT03313778	Active, not recruiting	aPD1	9
				II	High-risk melanoma		Intramuscular	NCT03897881	Recruiting	aPD1	9
				III	High-risk melanoma		Intramuscular	NCT05933577	Recruiting	aPD1	9

## Data Availability

Not applicable.

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
