# Peer review of "Personalized Cancer Vaccines Go Viral: Viral Vectors in the Era of Personalized Immunotherapy of Cancer"

_ijms, 2023, doi:10.3390/ijms242316591_

Round 1

Reviewer 1 Report

Comments and Suggestions for Authors

This is a very well written and informative review focusing on viral vectors and personalized vaccines for the immunotherapy of cancer.  Core concepts are clearly identified and characterized and the text is enhanced with well-organized tables of specific products currently in clinical trial as examples of this technology.  Figure 1 is also very helpful in describing the key steps for the successful production of these vaccines.  This reviewer has high enthusiasm for its publication but has a few comments that should be addressed in the text to clarify some issues, including:

1. To get a clearer picture of the production process, can the authors address the failure rate of specific vaccines and what are the mechanisms of failure.  In particular, if these vaccines do not induce sufficient memory responses, then tumors will regrow perhaps due to lack of target or incomplete memory induction.  

2. The other key point that needs to be addressed is that if tumors are HLA negative upon NGS examination, they should not be included in clinical trials since this type of vaccine depends on the presence of HLA I and II expressed by the tumors to present these neoantigens.  Do any of existing trials exclude HLA negative cases, and if not why?  Regarding HLA expression, if the reason for their inexpression is due to loss of chromosomal genes by deletion, then vaccine technology will not work.  If HLA loss is by epigenetic causes and can be re-expressed with interferon gamma/IL-12 pre-treatment, are any trials distinguishing the type of loss of HLA in their cases and can patients with "soft" loses, be pre-treated to upregulate HLA before vaccine administration.  Please comment.

3.  Cold tumors also need to be better characterized as to whether the lack of immunogenicity is due to either/or loss of HLA or high suppressor cell content in the periphery (adenosine secretion by Tregs) or by MDSC in the tumor microenvironment.  It might be useful to have a short paragraph describing why this is important to know in the context of selecting patients for vaccine trials.

Reviewer 2 Report

Comments and Suggestions for Authors

1. The theme of the review is well appreciated. The contents and their presentation flow smoothly. However, as the title of the review is broader, the review may be complete if there is a separate table showing  viral vectors including others, such as lentiviral and AAV vectors. Despite the focus on the vectors reviewed, providing comprehensive viral vector related information will satisfy reader curiosity and improve focus on the theme.

2. There is no mention of the final product release criteria or tests performed to that end, across the different platforms. Although TAT is discussed, other possible reasons of failure while using each of these platforms should also be discussed, atleast in brief. 

3. Table 1 presents only industry sponsored trials- no academic trial is mentioned. Either these should be included to broaden the scope of the information or explained in text that no such academic trials are being done.

4. Table 1 also fails to mention the status of these trials - whether they are ongoing or completed.

5. English language editing is mandatory as there are several grammatical and spelling errors present in the manuscript. This will certainly improve the clarity of the review.

Comments on the Quality of English Language

There are several error with regard to spelling, punctuation and grammar.

For e.g in the sentences 277-278 -" The capability of these viral vectors to target multiple neoantigens simultaneously is  key to fight tumor antigen heterogeneity " 

the apt word instead of "fight" would be "addressing". 

Table 1 carries the spelling "Intramuscolar" instead of intramuscular.

Hence thorough language editing will improve clarity of the manuscript.
